# Systematically constructing the likelihood for boosted $H \to gg$ decays

● Andrew J. Larkoski

American Physical Society, Hauppauge, New York 11788, USA

larkoski@aps.org

## Abstract

We study the binary discrimination problem of identification of boosted $H \to gg$ decays from massive QCD jets in a systematic expansion in the strong coupling. Though this decay mode of the Higgs is unlikely to be discovered at the LHC, we analytically demonstrate several features of the likelihood ratio for this problem through explicit analysis of signal and background matrix elements. Through leading-order, we prove that by imposing a constraint on the jet mass and measuring the energy fraction of the softer subjet an improvement of signal to background ratio that is independent of the kinematics of the jets at high boosts can be obtained, and is approximately equal to the inverse of the strong coupling evaluated at the Higgs mass. At next-to-leading order, we construct a powerful discrimination observable through a sort of anomaly detection approach by simply inverting the next-to-leading order $H \to gg$ matrix element with soft gluon emission, which is naturally infrared and collinear safe. Our analytic conclusions are validated in simulated data from all-purpose event generators and subsequent parton showering and demonstrate that the signal-to-background ratio can be improved by a factor of several hundred at high, but accessible, jet energies at the LHC.

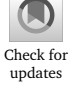

# 1  Introduction

The Higgs boson dominantly decays to ultimately observable hadrons [1] and so a detailed study of the Higgs requires identification of and discrimination from massive jets from quantum chromodynamics (QCD) that can fake a Higgs signature. Most of the time, the Higgs decays directly to bottom quarks, which has now been observed by the ATLAS and CMS experiments at the Large Hadron Collider (LHC) [2,3]. Much of the remaining decay width of the Higgs is through decays to off-shell electroweak bosons that subsequently decay to hadrons, but nearly 10% of the time, the Higgs also decays to gluons, through a top quark loop. The coupling of the Higgs to gluons is indirectly established through its dominant production mechanism at a hadron collider, gluon-gluon fusion, but direct observation of its decay to gluons would enable a probe of this coupling in a way that is significantly less sensitive to parton distribution functions.

However, identification of $H \to gg$ decays remains likely out of reach of the LHC because the QCD backgrounds are simply too overwhelming and zeroth-order methods for reducing the background, like bottom hadron tagging for the case of $H \to b\bar{b}$ decays, are not relevant. Nevertheless, identification of $H \to gg$ decays is an interesting academic exercise that allows for analytic study directly from appropriate matrix elements and subsequent validation and comparison to results from simulated data. This can then correspondingly be a testing and validation ground for machine learning methods of this problem, to ensure that any machine applied to this problem [4–23] is sensitive to at least the physics that can be identified and studied analytically.

To this goal, we will focus on discrimination of highly-boosted $H \to gg$ decays from massive jets initiated by light QCD partons. We will employ two guiding principles, namely, the Neyman-Pearson lemma [24] that the optimal discrimination observable is the likelihood ratio and infrared and collinear (IRC) safety whereby the distribution of an IRC safe observable can be calculated order-by-order in the strong coupling. We are therefore able to systematically construct the likelihood ratio as an expansion in the strong coupling and calculate discrimination metrics like the receiver operating characteristic curve, and correspondingly clearly identify the physics distinctions between signal and background. Working in the highly-boosted limit means that we can employ the relatively simple and universal collinear splitting functions [25–29] as appropriate matrix elements, especially for the background.

At leading-order, we demonstrate that a jet mass constraint, to ensure any possibility to be consistent with the on-shell decay of the Higgs, and measurement of the momentum fraction of the softer of the two leading subjets result in an improvement of the signal-to-background ratio that is independent of the jet's kinematics and approximately the inverse of the strong coupling evaluated at the Higgs mass, $1/\alpha_s \sim 10$. Beyond leading-order, the larger dimensionality of phase space and proliferation of background final states makes construction of the explicit form of the likelihood a bit tricky. Because of this, we instead approach the problem from the perspective of anomaly detection, see, e.g., Refs. [30–34], focusing on discrimination of signal from not-signal. Signal events are likely to populate phase space regions where the matrix element is large, and therefore are likely to populate regions where the inverse of the matrix element is small. Further, the inverse of the matrix element for $H \to gg$ decay at next-to-

leading order vanishes in the limit that the emitted gluon goes soft or becomes collinear to an initial decay product. As such, the inverse of the matrix element is naturally IRC safe itself, and suggests a more general anomaly detection observable construction that is amenable to theoretical calculations.

With results at next-to-leading order, we demonstrate that perfect discrimination is possible in the infinite jet energy limit, in which the problem reduces to discrimination of a color-singlet, $H \to gg$ decay, from color triplet or color octet jets of QCD. At high, but accessible jet energies at the LHC, we demonstrate that the signal-to-background ratio can be improved by a factor of several hundred, exclusively from measuring the jet's mass and the inverse of the $H \to ggg$ matrix element. An analysis that is sensitive to all particles and their momenta in the jets will necessarily be more powerful, but this simple baseline, founded solidly in the matrix elements of signal and background, provides a concrete ground reference for interpretation of what a machine is learning for this problem.

The outline of this paper is as follows. In Sec. 2, we present the simulated data samples that we analyze in concert with our theoretical analysis. In Sec. 3, we present our minimal assumptions for our analysis; namely, working to leading-order in the ratio of the Higgs mass to the jet's transverse momentum, and requiring that the mass of the jet is in a window about the Higgs mass. We compare analytic predictions for the fraction of background events that pass the mass cut to the corresponding fraction in simulated data and find good agreement, especially at high transverse momenta. In Sec. 4, we construct the likelihood ratio on jets at leading order with these assumptions, and show that the optimal discriminant is the energy fraction of the softer of the two leading subjets itself. We demonstrate quantitative agreement between predicted discrimination power and that from simulated data to this order. In Sec. 5, we study the problem at next-to-leading order and construct powerful discrimination observables from taking the difference of signal and background matrix elements or, even simpler, just inverting the signal matrix element at this order. These observables perform extremely well on simulated data, producing improvements of signal-to-background ratios of factors of hundreds. We conclude in Sec. 6, summarizing the results presented here, and looking forward to implementation of the general lessons into machine learning architectures.

## 2  Simulated data sample

Throughout this paper, we will organize our analysis in a systematic expansion in the strong coupling. We will present our theoretical analyses and their consequences and then immediately validate and test them in simulated data. Our theoretical analyses will be presented in the subsequent sections, but here, we present the simulated data samples that we study in more and more detail in the following. Leading-order $pp \to ZH$, with the Higgs subsequently decayed to gluons $H \to gg$, and $pp \to Z+$ jet events at the 13 TeV LHC were generated in MadGraph v3.6.0 [35]. The $Z$ boson was forced to decay to neutrinos. The decay of the Higgs to gluons in MadGraph uses the infinite top mass approximation. Events were then showered and hadronized in Pythia v8.306 [36] with default settings, and all particles except neutrinos were recorded for further analysis. Events were then clustered into jets with FastJet v3.4.0 [37]. Only the hardest anti-$k_T$ jet [38] with radius $R = 0.5$ and pseudorapidity less than 2.5 is then analyzed. This jet radius is smaller than typical radii used in experiment for heavy jet identification and correspondingly means that we can only consider jet transverse momenta above about $2m_H/R \sim 500$ GeV. A smaller radius does reduce the effect of contamination radiation, and we will mostly be interested in the large-boost jet tagging performance. In our analytic expressions, we will often refer to angles between pairs of particles as $\theta$, but on simulated jets

at a hadron collider, this angle is replaced by the longitudinal boost invariant distance,

$$\theta^2 \to \Delta\eta^2 + \Delta\phi^2\,, \tag{1}$$

where $\Delta\eta$ is the difference of the pseudorapidity of the particles and $\Delta\phi$ is the difference of azimuthal angles about the beam.

We will exclusively use the particles in the jets, and the relationships amongst them, for classification of boosted $H \to gg$ decays from massive jets initiated by QCD partons. We further additionally restrict to possible IRC safe information, which means that the observables on the jets we consider are only sensitive to the flow of energy within the jet, and not to individual particle identification. This does limit this analysis somewhat, because the background will contain a significant amount of jets initiated by light quarks, but will correspondingly be theoretically unambiguous and well-defined and further very robust to details and idiosyncrasies of simulated data. We further assume the narrow-width approximation throughout this analysis. This is explicitly used in event generation, and implicit in our definition of signal and background categories, so that there is no mixing of off-shell Higgs bosons with QCD jets at the amplitude level.

We will focus our analysis on the highly-boosted regime, where the characteristic transverse momentum to the beam of the jets is much larger than their mass, $p_\perp/m_H \ll 1$. At this level of event generation, the ratio of inclusive cross sections of $ZH$ versus $Z+$ jet production for jets with transverse momenta about 1 TeV is about 700:

$$\left.\frac{\sigma_{pp\to Zj}}{\sigma_{pp\to ZH}}\right|_{p_\perp > 1\text{ TeV}} \sim 700\,, \tag{2}$$

as calculated at leading-order in MadGraph. The Higgs decays to gluons about 10% of the time, so the initial background-to-signal ratio of jets that we consider is very large:

$$\left.\frac{\sigma_{pp\to Zj}}{\sigma_{pp\to Z(H\to gg)}}\right|_{p_\perp > 1\text{ TeV}} \sim 7000\,. \tag{3}$$

We will work through three orders of systematic approximations to reduce this as much as possible within this analysis.

## 3 Nominal assumptions at zeroth order

The first assumption that we employ is that for the jet to possibly be a Higgs, its mass must be consistent with that of the Higgs, $m_J \sim m_H \sim 125$ GeV. The leading-order contribution to a non-zero jet mass is for a jet with two particles. On signal jets, boosted $H \to gg$ decays, and with the narrow width approximation, a mass constraint about the Higgs mass is essentially always satisfied. On background jets, however, this constraint is non-trivial, but we can calculate its effect. We will work in the collinear limit to leading power in the ratio of the jet or Higgs mass to the jet transverse momentum, $m_H/p_\perp \ll 1$, and therefore can calculate the jet mass with collinear splitting functions. QCD jets at leading order are initiated either by light quarks or gluons, and we can calculate the relative probability as compared to inclusive production on quark and gluon jets separately.

## 3.1 Theoretical analysis

The probability that the mass of a quark-initiated jet is in a window of size $dm_H^2$ about the Higgs mass can be calculated from

$$p(\text{Higgs mass}|q) = \frac{\alpha_s C_F}{2\pi} \frac{dm_H^2}{m_H^2} \int dz \, \frac{1 + (1-z)^2}{z} \Theta\left(R^2 - \frac{m_H^2}{z(1-z)p_\perp^2}\right), \qquad (4)$$

where $C_F$ is the fundamental quadratic Casimir of SU(3) color and takes the value $C_F = 4/3$ in QCD. At far right, the $\Theta$-function restricts both emissions to lie within the jet of radius $R$ and transverse momentum $p_\perp$, and $z$ is the energy fraction of the gluon in $q \to qg$ fragmentation. The integral over the energy fraction can be evaluated, and we find

$$p(\text{Higgs mass}|q) = \frac{\alpha_s C_F}{\pi} \frac{dm_H^2}{m_H^2} \left( \log \frac{R^2 p_\perp^2}{m_H^2} - \frac{3}{4} \right), \qquad (5)$$

to leading power in $m_H/p_\perp$. Terms suppressed by $m_H/p_\perp$ are not universal anyway, and depend on the particular process, colliding partons, or phase space constraints on the jets, so we cannot make a clean theoretical prediction for them.

For gluon-initiated jets, the corresponding calculation is

$$p(\text{Higgs mass}|g) \qquad\qquad\qquad\qquad\qquad\qquad\qquad\qquad\qquad\qquad (6)$$
$$= \frac{\alpha_s}{2\pi} \frac{dm_H^2}{m_H^2} \int dz \left[ C_A \left( \frac{z}{1-z} + \frac{1-z}{z} + z(1-z) \right) + n_f T_R \left( z^2 + (1-z)^2 \right) \right] \Theta\left(R^2 - \frac{m_H^2}{z(1-z)E^2}\right)$$
$$= \frac{\alpha_s}{\pi} \frac{dm_H^2}{m_H^2} \left( C_A \log \frac{R^2 p_\perp^2}{m_H^2} - \frac{11}{12} C_A + \frac{n_f T_R}{3} \right),$$

where $C_A = 3$ is the adjoint quadratic Casimir of SU(3) color, $n_f$ is the number of active/light quarks, and $T_R = 1/2$ is the normalization of the fundamental representation of SU(3).

Then, to fit this with inclusive jet production, we would need to sum these quark and gluon results together, multiplied by their respective probabilities for production,

$$p(\text{Higgs mass}|\text{QCD}) = p(\text{Higgs mass}|q)p(q) + p(\text{Higgs mass}|g)p(g). \qquad (7)$$

The individual quark and gluon production probabilities, $p(q)$ and $p(g)$, depend in detail on phase space cuts, parton distributions, and other collider-dependent factors, so can change depending on the process we are considering. However, we can nevertheless bound any possible probability noting that $C_F < C_A$ and so

$$p(\text{Higgs mass}|q) \leq p(\text{Higgs mass}|\text{QCD}) \leq p(\text{Higgs mass}|g). \qquad (8)$$

We will use these bounds when comparing to simulation.

## 3.2 Comparison with simulation

On our simulated data, we then apply a mass cut on the jets. We make a fixed mass cut window of $m_J \in [110, 150]$ GeV and consider jets with transverse momentum in four different bins: $p_\perp > 500, 1000, 1500,$ and $2000$ GeV, to study the dependence on transverse momentum. With this choice of mass cut window, note that the corresponding ratio in our analytic calculation is

$$\frac{dm_H^2}{m_H^2} = 2\frac{dm_H}{m_H} \approx 2\frac{40}{125} = 0.64, \qquad (9)$$

Table 1: Table of total events that pass inclusive, jet level restrictions and that pass the mass cut about the Higgs mass on signal and background events. Fractions that pass the mass cut are also displayed.

|  | Signal |  | Background |  |
|---|---|---|---|---|
| $p_\perp > 500$ GeV | 34878 events | 1 | 343192 events | 1 |
| $m_J \in [110, 150]$ GeV | 23077 events | 0.662 | 18437 events | 0.0537 |
| $p_\perp > 1000$ GeV | 28941 events | 1 | 257357 events | 1 |
| $m_J \in [110, 150]$ GeV | 25564 events | 0.883 | 32399 events | 0.126 |
| $p_\perp > 1500$ GeV | 21176 events | 1 | 189376 events | 1 |
| $m_J \in [110, 150]$ GeV | 19530 events | 0.918 | 30105 events | 0.159 |
| $p_\perp > 2000$ GeV | 15065 events | 1 | 132002 events | 1 |
| $m_J \in [110, 150]$ GeV | 13947 events | 0.926 | 22385 events | 0.170 |

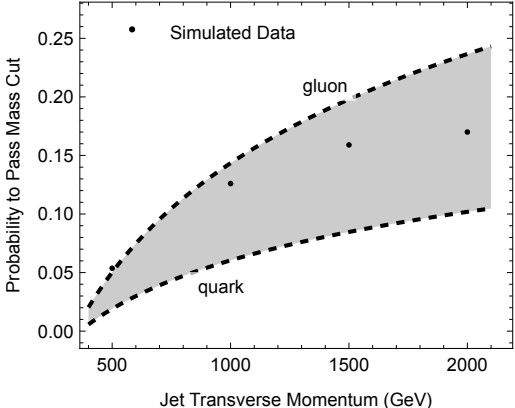

Figure 1: Plot of the predicted fractions of quark and gluon jets that pass the Higgs mass cuts, Eqs. (5) and (6), as a function of jet transverse momentum. Results from simulated data (as listed in Table 1) are displayed as the black dots. The grey region illustrates where a general sample as a linear combinations of quark and gluon jets should lie.

which we appropriately include in evaluation of the expressions above. First, however, on these data, we display the total number and fraction of events that pass the jet-level and mass cuts in Table 1. As expected, the mass constraint on signal events has little effect, and actually becomes more efficient at higher transverse momentum, where a greater fraction of the boosted Higgs have all decay products within the jet. A significant fraction of background jets are removed by the mass cut, but the fraction that passes also increases as transverse momentum increases.

This latter observation is expected from our theoretical predictions of Eqs. (5) and (6) which exhibit logarithmic sensitivity to the transverse momentum cut. To illustrate this, we plot the theoretical quark and gluon bounds and the corresponding fractions from simulation that pass the mass cut. To evaluate the probabilities, we use $\alpha_s = 0.11$, which is approximately the value at the scale of the Higgs mass, and the result is displayed in Fig. 1. Simulated data fractions, black dots in the plot, lie within the expected bounds at sufficiently high transverse momenta. The 500 GeV data point lies near (and perhaps above) the upper bound, but we note that there, the relevant ratio of scales is

$$\frac{m_H^2}{R^2 p_\perp^2} \sim \frac{125^2}{0.5^2 \cdot 500^2} \approx 0.25\,, \tag{10}$$

which is not especially small, so we expect that power corrections may be important. However, by the highest transverse momentum, this scale is reduced to

$$\frac{m_H^2}{R^2 p_\perp^2} \sim \frac{125^2}{0.5^2 \cdot 2000^2} \approx 0.016,\tag{11}$$

and so power corrections are expected to be minimal, and the leading-power expressions should dominate.

# 4 Likelihood at leading order

From these nominal results, we now move to implementing more exclusive cuts on the jets at leading order in the strong coupling. In our collinear, highly-boosted approximation, phase space for the jets is one-dimensional after imposing the mass cut, so we will be able to present compact, analytic expressions for all results. Our guiding principle will be the Neyman-Pearson lemma [24], for which the optimal discrimination observable is monotonic in the likelihood ratio, or the ratio of signal and background probability distributions. Our theory goal will then be to calculate the distribution of the likelihood and then to compare the subsequent predicted discrimination power to the outcome of simulation.

## 4.1 Theoretical analysis

Our first step to this goal is to calculate the probability distributions on phase space on signal and background jets at leading order in the strong coupling. On signal, $H \to gg$ jets, we note that as a spin-0 scalar decaying to two massless particles, the squared matrix element for Higgs decay is Lorentz invariant and flat on phase space. We will take the relevant phase space variable to be the momentum fraction of the softer particle, $z = \min[z_1, z_2]$, and so the normalized probability distribution on phase space for signal jets is

$$p_H(z) = 2,\tag{12}$$

where $z \in [0, 1/2]$. Because there is a finite jet radius, the minimum value of $z$ is technically $m_H^2/(R^2 p_\perp^2)$, but because the Higgs distribution is smooth at small $z$, this region is power-suppressed, and so, in the high-boost limit, we can safely ignore it.

For background, we will consider quark- and gluon-initiated jets separately, and later comment on their relationship. For quark jets, the distribution of the softer particle can be calculated from

$$p_q(z) \propto \frac{\alpha_s C_F}{2\pi} \frac{dm_H^2}{m_H^2} \int dz' \frac{1 + (1-z')^2}{z'} \Theta\left(R^2 - \frac{m_H^2}{z'(1-z')p_\perp^2}\right)\tag{13}$$

$$\times \left[\Theta(1/2 - z')\delta(z - z') + \Theta(z' - 1/2)\delta(z - (1-z'))\right]$$

$$= \frac{\alpha_s C_F}{2\pi} \frac{dm_H^2}{m_H^2} \left(\frac{1 + (1-z)^2}{z} + \frac{1 + z^2}{1-z}\right) \Theta\left(z - \frac{m_H^2}{R^2 p_\perp^2}\right).$$

Because the energy fraction distribution diverges at small $z$, the jet restriction is important, and cannot be ignored. Earlier, we had actually calculated the normalization factor for this probability distribution, and so the unit normalized distribution is

$$p_q(z) = \frac{1}{2\log\frac{R^2 p_\perp^2}{m_H^2} - \frac{3}{2}} \left(\frac{1 + (1-z)^2}{z} + \frac{1 + z^2}{1-z}\right),\tag{14}$$

on

$$z \in \left[ \frac{m_H^2}{R^2 p_\perp^2}, \frac{1}{2} \right], \tag{15}$$

and to leading power in $m_H/p_\perp \ll 1$. The unit-normalized gluon energy fraction distribution can be calculated similarly, which we just display the result here:

$$p_g(z) = \frac{1}{2C_A \log \frac{R^2 p_\perp^2}{m_H^2} - \frac{11}{6}C_A + \frac{2}{3}n_f T_R} \left[ C_A \left( \frac{z}{1-z} + \frac{1-z}{z} + z(1-z) \right) + n_f T_R \left( z^2 + (1-z)^2 \right) \right]. \tag{16}$$

These energy fraction distributions have been proposed and calculated in the context of jet grooming [39, 40], corresponding to the first emission that passes the grooming criteria. In the case at hand, these distributions are IRC safe because we have constrained emissions on the jet by the measured mass, and so an exactly collinear splitting is not allowed. By contrast, the observable proposed in the grooming context is not IRC safe, but is Sudakov safe [41], because a first emission in the collinear region is exponentially suppressed by the approximate scale invariance of QCD.

We also note that these energy fraction distributions on quark and gluon jets are numerically very close to one another. Indeed, the distributions would be exactly the same if QCD were supersymmetric, for which $C_F = C_A = n_f$ [42, 43]. Therefore, even when linear combinations of quark and gluon jets are constructed to represent the composition of the background event ensemble, there is little dependence on the precise quark and gluon fractions. As such, we will just study the distribution of Eq. (14) in the following as examplar for all jets, and further, which has no parameters, except the kinematic constraints on the jet.

With this set-up, note that the likelihood ratio is

$$\mathcal{L} = \frac{p_q(z)}{p_H(z)} = \frac{1}{4 \log \frac{R^2 p_\perp^2}{m_H^2} - 3} \left( \frac{1 + (1-z)^2}{z} + \frac{1 + z^2}{1-z} \right), \tag{17}$$

which is monotonically-decreasing in energy fraction $z$, and therefore the energy fraction itself is equivalently the optimal discrimination observable, with these assumptions. Then, from the distributions in $z$ exclusively, we can calculate the signal-versus-background plot, or receiver operating characteristic (ROC) curve. The cumulative distribution $\Sigma(z)$ for either signal or background at this order is defined as

$$\Sigma(z) = \int_0^z dz' \, p(z'), \tag{18}$$

where $p(z)$ is the corresponding probability distribution of the energy fraction $z$. We then define the function of cumulative distributions in $z$, where

$$F(x) = \Sigma_q \left( \Sigma_H^{-1}(x) \right), \tag{19}$$

where $\Sigma_q(z)$ is the cumulative distribution of the energy fraction on background, and $\Sigma_H^{-1}(x)$ is the inverse cumulative distribution on signal, at quantile $x$. Note that the likelihood is large at small $z$, corresponding to the background-rich region. However, it is often more desirable to plot the ROC curve with the signal efficiency on the abscissa, which requires a simple change of variables from $F(x)$. That is, the ROC curve we will consider is

$$\text{ROC}(x) = 1 - F(1-x) = 1 - \Sigma_q \left( \Sigma_H^{-1}(1-x) \right). \tag{20}$$

To construct this ROC curve, we need the cumulative distributions. These are

$$\Sigma_H(z) = 2z \,, \tag{21}$$

$$\Sigma_q(z) = \frac{1}{2\log\frac{R^2 p_\perp^2}{m_H^2} - \frac{3}{2}} \left( 2\log\frac{zR^2 p_\perp^2}{m_H^2} - 2\log(1-z) - 3z \right). \tag{22}$$

The auxiliary function $F(x)$ is then

$$F(x) = \Sigma_q\left(\Sigma_H^{-1}(x)\right) = \frac{1}{2\log\frac{R^2 p_\perp^2}{m_H^2} - \frac{3}{2}} \left( 2\log\frac{xR^2 p_\perp^2}{2m_H^2} - 2\log\left(1-\frac{x}{2}\right) - \frac{3}{2}x \right), \tag{23}$$

and so the ROC curve is

$$\text{ROC}(x) = \frac{\log\frac{1+x}{1-x} - \frac{3}{4}x}{\log\frac{R^2 p_\perp^2}{m_H^2} - \frac{3}{4}}. \tag{24}$$

From the ROC curve, another useful discrimination metric is the area under the ROC curve, or, the AUC. We can calculate this area simply, where

$$\text{AUC} = \int dx\, \text{ROC}(x) = \frac{16\log 2 - 3}{8\log\frac{R^2 p_\perp^2}{m_H^2} - 6}, \tag{25}$$

to leading power in $m_H/p_\perp$. This does predict arbitrarily good discrimination at sufficiently high transverse momentum; however, the approach is logarithmically slow. Further, power corrections seem to be rather important. For example, random jet selection, the worst possible discriminant, has an AUC of 0.5, but this expression predicts an AUC of 0.5 for $R = 0.5$ and $p_\perp = 1000$ GeV. So, direct analytic comparisons to results from simulated data will be limited, but we still expect and predict that the energy fraction is a good discriminant at sufficiently high jet energies, nonetheless.

To our overall goal of reducing the inclusive background to signal cross section ratio to hopefully discover $H \to gg$ decays, another interesting distribution to consider is the ratio of signal to background fractions, as a function of the signal fraction. From the ROC curve, this is

$$\frac{S}{B}(x) = \frac{x}{\text{ROC}(x)} = \frac{x}{\log\frac{1+x}{1-x} - \frac{3}{4}x} \left( \log\frac{R^2 p_\perp^2}{m_H^2} - \frac{3}{4} \right). \tag{26}$$

What is especially interesting about this is that this is monotonically-decreasing in signal fraction $x$, and the maximum at $x = 0$ is

$$\lim_{x \to 0} \frac{S}{B}(x) = \frac{4}{5}\log\frac{R^2 p_\perp^2}{m_H^2} - \frac{3}{5}. \tag{27}$$

Recall that, for quark-initiated jet for concreteness, the fraction of background events that passed the mass cut was

$$p(\text{Higgs mass}|q) = \frac{\alpha_s C_F}{\pi} \frac{dm_H^2}{m_H^2} \left( \log\frac{R^2 p_\perp^2}{m_H^2} - \frac{3}{4} \right), \tag{28}$$

while effectively all signal Higgs jets pass the mass cut. Then, with this result for the modification to the signal versus background ratio from a cut on the energy fraction, the fraction of

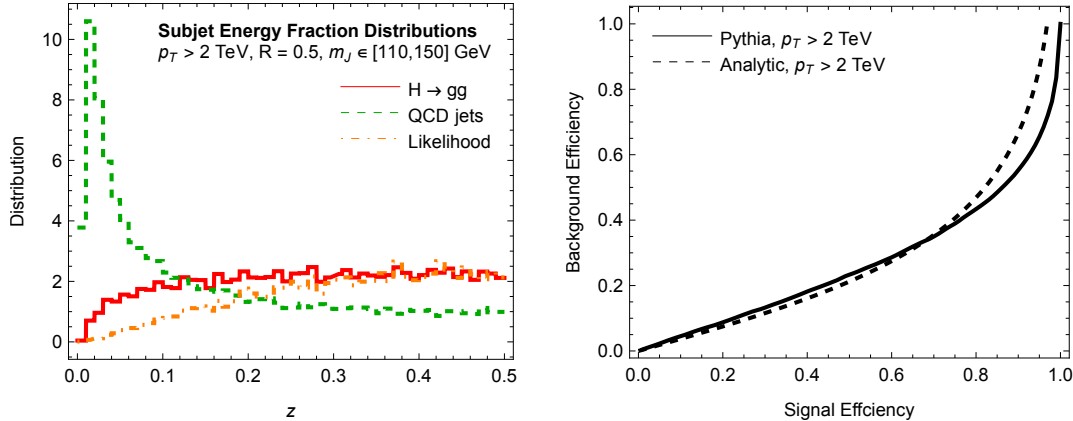

Figure 2: Left: Distributions of the subjet momentum fraction $z$ on Higgs jets (red) and QCD jets (dashed green) in the simulated data sample with jets with $p_\perp > 2$ TeV. Also plotted is the ratio of the Higgs to QCD distributions (dot-dashed orange). Right: ROC curves as determined from the momentum fraction distributions in the $p_\perp > 2$ TeV simulated jets sample (solid) and the analytic expression of Eq. (24) (dashed).

background events that pass the mass and energy fraction cuts versus that of signal events is roughly

$$\Delta \frac{B}{S} \approx \frac{p(\text{Higgs mass}|q)}{\lim_{x \to 0} \frac{S}{B}(x)} \approx \frac{5}{4} \frac{\alpha_s C_F}{\pi} \frac{dm_H^2}{m_H^2},\qquad(29)$$

approximately independent of the kinematics of the jets. Thus, from mass and subjet energy fraction cuts, the background to signal cross section ratio of Eq. (3) can be reduced by roughly a factor of the coupling evaluated at the scale of the Higgs mass, $\alpha_s(m_H) \sim 0.1$.

## 4.2 Comparison with simulation

On the jet samples that are already selected to pass the mass constraints, to define the soft subjet energy fraction, we do the following. The jets are reclustered with the exclusive $k_T$ algorithm [44, 45] with Winner-Take-All recombination [46–48] into two subjets. Then, from these two subjets, we define the energy fraction $z$ as

$$z \equiv \frac{\min[p_{\perp,1}, p_{\perp,2}]}{p_\perp},\qquad(30)$$

where $p_\perp$ is the total jet transverse momentum, and $p_{\perp,1}, p_{\perp,2}$ are the transverse momenta of the two subjets. Winner-Take-All recombination is not important at this stage, but will be in the following section as we refine the likelihood.

Given these data, we then analyze the energy fraction distributions in simulation. At left in Fig. 2, we plot the signal and background energy fraction distributions in the $p_\perp > 2000$ GeV sample. The distributions have the expected form: very flat for signal, and highly peaked at small $z$ on background jets. One thing to note that is beyond our analytic study is that the energy fraction distribution on signal does not sharply turn off at the kinematic minimum, but rather tails off smoothly at small values of $z$. This is due to particles produced in the subsequent shower of the Higgs decay products that are not captured in the jet. Importantly, note that in our analytic calculation, this is effectively a higher-order effect, in which additional radiated emissions from the subjets are sculpted by the jet finding. Rather amusingly, this effect actually decreases the overlap between signal and background, improving discrimination

power with this observable. Also on this plot, we display the ratio of signal to background distributions (which makes displaying all curves on this plot easier). As predicted, up to statisical fluctuations in data, this ratio is monotonic in the energy fraction $z$, and so the energy fraction itself is the optimal discriminant.

At right in Fig. 2, we plot the ROC curve from simulated data in the same event class, and compare to the analytic expression from Eq. (24). Rather impressive agreement between simulation and analytics are observed over most of the signal efficiency range, and then there is some divergence at very high signal efficiencies (corresponding to the region at small values of $z$). This is expected, given the distributions plotted at left, but nevertheless validate that our analytic results do indeed largely describe the physics of simulation to this order. At smaller jet transverse momentum, this sculpting effect at small energy fraction $z$ becomes more pronounced, and correspondingly more challenging to predict, which is why we focused here on jets in the highest transverse momentum bin.

## 5 Likelihood at next-to-leading order and beyond

We continue our analysis, moving on to sensitivity to the leading emission off of the initial two hard prongs in the jets. A theoretical analysis at this order could be performed as in Ref. [49], in which the likelihood is constructed at next-to-leading order, and then subsequently the ROC curve is calculated that would represent the theoretically-optimal discrimination performance to this accuracy. However, with our goal of concrete observables that can be implemented in an experimental analysis, we will take a slightly different direction. Instead, we will study the dominant contributions to signal and background jets at next-to-leading order and then construct observables sensitive to emissions at this order and beyond, and then construct the likelihood on this reduced space. We will then validate that the ultimate observables we construct from this process do indeed perform well on simulated data.

### 5.1 Theoretical analysis

Our first step of the theoretical analysis at this order is to identify signal and background jets. Signal is obvious, a study at next-to-leading order of boosted $H \to gg$ decays. Earlier, background had been any jet initiated by light QCD partons, quarks or gluons, and this results in an abundance of configurations to consider. So, instead, we narrow our analysis to just those background jets that look as close as possible to signal jets, namely, massive jets from $g \to gg$ fragmentation. Additionally, starting at next-to-leading order, emissions are first sensitive to the flow of color in the jet. In particular, soft gluon emissions are dominantly sensitive to color flow. We will therefore focus on the leading structure of soft gluon emissions from $H \to gg$ versus $g \to gg$ jets, and construct observables sensitive to the distinct structures of signal and background.

To do this, we need to calculate distributions on these jets with soft gluon emission. This requires knowledge of the phase space and squared matrix element for soft gluon emission. For the squared matrix element, the leading contribution in the soft limit follows from the well-known eikonal amplitude [50–53], where

$$\frac{p(\Pi_3)}{p(\Pi_2)} = -(4\pi)^2 \frac{\alpha_s}{2\pi} \sum_{i,j} \mathbf{T}_i \cdot \mathbf{T}_j \frac{s_{ij}}{s_{ik} s_{jk}} + \dots \tag{31}$$

Here, $p(\Pi_3)$ is the distribution of three particles on phase space $\Pi_3$, $\mathbf{T}_i$ is the color matrix of hard particle $i$, $s_{ij}$ is the invariant mass of particles $i$ and $j$, and the ellipses represents terms that are subleading in the soft limit of gluon $k$. A soft particle cannot affect the kinematics of

hard particles to leading power, and so the phase space for a soft emission is conditioned on the kinematics of the hard particles. Single soft gluon phase space on a collinear jet can be expressed as

$$d\Pi = \frac{2p_\perp^2}{(4\pi)^3} \, d\theta^2 \, d\phi \, z \, dz \,, \tag{32}$$

where $p_\perp$ is the transverse momentum of the jet, $\theta$ is an angle from a hard particle, $\phi$ is the azimuthal angle about a hard particle in the jet, and $z$ is the energy fraction of the soft gluon.

With these expressions, the squared matrix element for soft gluon emission from boosted $H \to gg$ can be expressed as

$$|\mathcal{M}^{(H)}|^2 = 2(4\pi)^2 \frac{\alpha_s C_A}{2\pi} \frac{1}{p_\perp^2} \frac{\theta_{12}^2}{z^2 \theta_{1k}^2 \theta_{2k}^2} = 2(4\pi)^2 \frac{\alpha_s C_A}{2\pi} \frac{1}{p_\perp^2} \frac{\theta_{12}^2}{z^2 \theta^2 \left(\theta_{12}^2 + \theta^2 - 2\theta\theta_{12}\cos\phi\right)} \,, \tag{33}$$

where, at right, we have expressed the squared matrix element in the phase space coordinates above, using the law of cosines and $\theta_{12}$ is the angle between the two leading subjets. The squared matrix element for soft gluon emission from collinear $g \to gg$ fragmentation is

$$\begin{aligned}
|\mathcal{M}^{(g)}|^2 &= (4\pi)^2 \frac{\alpha_s C_A}{2\pi} \frac{1}{p_\perp^2} \frac{\theta_{1k}^2 + \theta_{2k}^2 + \theta_{12}^2}{z^2 \theta_{1k}^2 \theta_{2k}^2} \\
&= 2(4\pi)^2 \frac{\alpha_s C_A}{2\pi} \frac{1}{p_\perp^2} \frac{\theta^2 + \theta_{12}^2 - \theta\theta_{12}\cos\phi}{z^2 \theta^2 \left(\theta_{12}^2 + \theta^2 - 2\theta\theta_{12}\cos\phi\right)} \,.
\end{aligned} \tag{34}$$

This squared matrix element also follows from the soft limit of the corresponding $g \to ggg$ splitting function [54, 55].

Now, naively, from these squared matrix elements, their ratio would be the likelihood we seek [56], but this is problematic because these matrix elements are not integrable on soft gluon phase space. Thus, they are not proper probability distributions, and as such the assumptions of the Neyman-Pearson lemma do not apply. Nevertheless, we will consider two resolutions to this apparent dilemma. First, considering these matrix elements as an $\mathcal{O}(\alpha_s)$ correction to the leading-order distributions we studied earlier, for consistency, we should only expand their total ratio to $\mathcal{O}(\alpha_s)$. Doing so results in the difference of squared matrix elements as the relevant contribution to the likelihood to this order [49]:

$$|\mathcal{M}^{(g)}|^2 - |\mathcal{M}^{(H)}|^2 = (4\pi)^2 \frac{\alpha_s C_A}{2\pi} \frac{1}{p_\perp^2} \frac{\theta_{1k}^2 + \theta_{2k}^2 - \theta_{12}^2}{z^2 \theta_{1k}^2 \theta_{2k}^2} \,. \tag{35}$$

Continuing with this approach is the way to calculate the complete likelihood and ROC curve on phase space through next-to-leading order.

This difference has an interesting structure that is sensitive to the geometry of the particles in the jet. Because the Higgs is a color singlet, it dominantly emits in the region between the two hard gluons, while the gluon is itself a color-octet, and so will emit at wide angles from the initial hard particles. This then suggests that the following observable, IRC safe and inclusive over additional emissions,

$$\mathcal{O}_{\text{NLO}} \equiv \frac{p_\perp^2}{m_H^2} \sum_k z_k \left(\theta_{1k}^2 + \theta_{2k}^2 - \theta_{12}^2\right) \,, \tag{36}$$

takes very different values on signal and background. Here, the sum is over all particles $k$ in the jet, and the direction of subjets 1 and 2 are defined by an appropriate reclustering algorithm. The overall factor of the jet $p_\perp$ and mass ensures that this observable is invariant to boosts

along the jet direction. A related color-sensitive observable was constructed for top tagging in Ref. [57].

The second observable we consider is constructed as follows. To identify the Higgs jets from background, whatever background it may be, we want to identify and remove those jets that only populate regions of phase space unlikely to have come from Higgs jets. This is a sort of anomaly detection where we remain agnostic as to the precise structure of background. Nevertheless, Higgs jets are unlikely to populate regions of phase space in which the matrix element is small; or, conversely, Higgs events are likely to populate those regions where the inverse of the matrix element is small. What is particularly interesting about this is that the kinematic dependence of the inverse of the Higgs soft gluon matrix element is IRC safe itself. So, summing over all emissions in the jet to be inclusive over additional structure, the second observable we consider is

$$d_2 \equiv \frac{p_\perp^2}{m_H^2} \sum_k z_k \frac{\theta_{1k}^2 \theta_{2k}^2}{\theta_{12}^2}. \tag{37}$$

We call this observable $d_2$ because of its similarity to the observable $D_2$ introduced in Ref. [58] from the energy correlation functions [59].

One can consider other functional forms of observables with various desired properties, but we will just stick with these two for concreteness, and some study of assumed functional dependence. Now, given these observables, we would like to calculate their distribution on signal and background events. Working in the soft limit and conditioning on the energy fraction of one of the hard subjets, $z$, the master formula for the distribution is

$$p(\mathcal{O}|z) = \int d\Pi \, |\mathcal{M}|^2 \, \delta(\mathcal{O} - \hat{\mathcal{O}}(\Pi)), \tag{38}$$

where $\hat{\mathcal{O}}(\Pi)$ is the functional form of the observable on phase space. We are being slightly sloppy with notation here, because fixed-order distributions, as considered here, are in general not probability densities, however, we won't need to worry about that yet.

While interpreting the result of Eq. (38) as a probability density is problematic, what is well-defined are moments of IRC safe observables. What we will consider here is the moment of an observable $\mathcal{O}$, conditioned on the minimum subjet momentum fraction $z$, where

$$\langle \mathcal{O}|z \rangle = \int d\Pi \, |\mathcal{M}|^2 \, \hat{\mathcal{O}}(\Pi). \tag{39}$$

This moment will correspondingly define the leading relationship between the observable and the momentum fraction, and, as such, be used to construct an observable formed from their combination that performs between than either individually for discrimination. This procedure results in the same functional relationships between observables that would also be found by the power counting approach of Ref. [58], but in some ways requires less thought because one just needs to evaluate an integral.

Starting with the observable $\mathcal{O}_{\text{NLO}}$ on Higgs jets, its moment is

$$\langle \mathcal{O}_{\text{NLO}}|z \rangle_H = \frac{2}{\pi} \frac{\alpha_s C_A}{2\pi} \frac{2p_\perp^2}{m_H^2} \int d\theta \, d\phi \, dz' \, \frac{\theta_{12}^2 (\theta - \theta_{12} \cos\phi)}{(\theta_{12}^2 + \theta^2 - 2\theta \theta_{12} \cos\phi)}. \tag{40}$$

In the gluon momentum fraction integral, we need to enforce that it is less than the momentum fraction of the soft subjet, $z$, to be actually soft. However, on signal jets, $z \sim 1$, and so this constraint does not introduce any new parametric scales, so can be ignored. This then integrates to

$$\langle \mathcal{O}_{\text{NLO}}|z \rangle_H = 4 \frac{\alpha_s C_A}{2\pi} \frac{\theta_{12}^2 p_\perp^2}{m_H^2} \log \frac{R^2}{\theta_{12}^2} \approx 4 \frac{\alpha_s C_A}{2\pi} \log \frac{p_\perp^2 R^2}{m_H^2}, \tag{41}$$

where, at right, we have used the parametric approximation that $\theta_{12}^2 p_\perp^2 \sim m_H^2$ on signal jets. This same moment on background jets is

$$
\begin{aligned}
\langle \mathcal{O}_{\text{NLO}} | z \rangle_g &= \frac{2}{\pi} \frac{\alpha_s C_A}{2\pi} \frac{2p_\perp^2}{m_H^2} \int d\theta \, d\phi \, dz' \frac{\left(\theta^2 + \theta_{12}^2 - \theta \theta_{12} \cos\phi\right)\left(\theta - \theta_{12} \cos\phi\right)}{\left(\theta_{12}^2 + \theta^2 - 2\theta \theta_{12} \cos\phi\right)} \\
&= 4 \frac{\alpha_s C_A}{2\pi} \frac{R^2 p_\perp^2}{m_H^2} z \left(1 - \frac{\theta_{12}^2}{2R^2} + \frac{\theta_{12}^2}{2R^2} \log \frac{R^2}{\theta_{12}^2}\right).
\end{aligned}
\tag{42}
$$

Now, for background, we must impose the non-trivial upper bound on the gluon momentum fraction $z'$ integral to be less than the softer subjet's momentum $z$ as, dominantly, $z \ll 1$.

From these results, we can make a few observations. First, on signal jets, the expectation value of the observable $\langle \mathcal{O}_{\text{NLO}} | z \rangle_H$ is approximately independent of the momentum fraction $z$. By contrast, on background jets, the expectation value scales approximately linear with $z$, where we note that $\theta_{12} \sim R$ on background. Further, this background moment is a linear combination of terms and is actually smaller than that of the signal. These observations suggests that a good discriminant observable on the $(\mathcal{O}_{\text{NLO}}, z)$ phase space is a linear function whose deviation from constant is determined by the value of $\mathcal{O}_{\text{NLO}}$, divided by the energy fraction $z$:

$$
\mathcal{L}_{\{\mathcal{O}_{\text{NLO}}, z\}} \simeq \frac{1 + \mathcal{O}_{\text{NLO}}}{z}.
\tag{43}
$$

We can continue with a similar analysis for the observable $d_2$, calculating its moment conditioned on the value of the soft subjet momentum fraction $z$. On signal jets, we have

$$
\langle d_2 | z \rangle_H = \frac{2}{\pi} \frac{\alpha_s C_A}{2\pi} \frac{p_\perp^2}{m_H^2} \int d\theta^2 \, d\phi \, dz' = 4 \frac{\alpha_s C_A}{2\pi} \frac{R^2 p_\perp^2}{m_H^2},
\tag{44}
$$

where again we use that $z \sim 1$ on signal. Note that the observable and squared matrix element cancel exactly, as constructed on signal jets, and result in a moment that scales like $p_\perp^2$. By contrast, the moment on background jets is

$$
\begin{aligned}
\langle d_2 | z \rangle_g &= \frac{2}{\pi} \frac{\alpha_s C_A}{2\pi} \frac{2p_\perp^2}{m_H^2} \int d\theta \, d\phi \, dz' \, \theta \, \frac{\theta^2 + \theta_{12}^2 - \theta \theta_{12} \cos\phi}{\theta_{12}^2} \\
&= 4 \frac{\alpha_s C_A}{2\pi} \frac{R^2 p_\perp^2}{m_H^2} \frac{z}{\theta_{12}^2} \left(\frac{R^2}{2} + \theta_{12}^2\right).
\end{aligned}
\tag{45}
$$

In this expression, the angular factor in parentheses is proportional to $R^2$, because the emission that sets the mass is emitted near the jet boundary on background. Through the mass constraint in the soft limit, $\theta_{12}^2 \propto 1/z$, and so this moment scales roughly like $z^2$. This then suggests that a good discriminant observable on the $(d_2, z)$ space is the ratio

$$
\mathcal{L}_{\{d_2, z\}} \simeq \frac{d_2}{z^2}.
\tag{46}
$$

We now present a parametric analysis of the discrimination power of the observable $\mathcal{L} = d_2/z^2$ in the soft gluon emission limit at next-to-leading order to illustrate and validate its efficacy. To do this, we will calculate the cumulative distributions of $\mathcal{L}$ on both signal and background jets, and illustrate that background jets especially have a distribution that strongly depends on the energy scale of the jet. We first construct the cumulative distribution for signal jets, which

can be expressed as

$$\Sigma_H(\mathcal{L}) = 1 \tag{47}$$

$$-\frac{\alpha_s C_A}{2\pi} \frac{4}{\pi} \int_0^{1/2} dz \int d\theta \, d\phi \, dz' \frac{\theta_{12}^2}{z'\theta \left(\theta_{12}^2 + \theta^2 - 2\theta\,\theta_{12}\cos\phi\right)} \Theta\left(\frac{p_\perp^2}{m_H^2} \frac{z'}{z^2} \frac{\theta^2\left(\theta_{12}^2 + \theta^2 - 2\theta\,\theta_{12}\cos\phi\right)}{\theta_{12}^2} - \mathcal{L}\right)$$

$$= 1 - \frac{\alpha_s C_A}{2\pi} \frac{4}{\pi} \int_0^{1/2} dz \int dx \, d\phi \, dz' \frac{1}{z'x\left(1 + x^2 - 2x\cos\phi\right)} \Theta\left(\frac{z'x^2\left(1 + x^2 - 2x\cos\phi\right)}{z^3(1-z)} - \mathcal{L}\right),$$

where to avoid the treatment of the singular region of phase space, we have used unitarity. Note that the integrand is convergent as the rescaled angle $x \to \infty$, and so enforcing a finite jet radius would only be a power correction, and so is negligible in the high-boost limit. Therefore, this cumulative distribution is purely a function of the observable value $\mathcal{L}$, with no dependence on the jet transverse momentum whatsoever.

By constrast, the background cumulative distribution for this observable is

$$\Sigma_g(\mathcal{L}) = 1 - \frac{\alpha_s C_A}{2\pi} \frac{2}{\pi} \frac{1}{2\log\frac{R^2 p_\perp^2}{m_H^2} - \frac{3}{2}} \int_{\frac{m_H^2}{R^2 p_\perp^2}}^{1/2} dz \left(\frac{1 + (1-z)^2}{z} + \frac{1+z^2}{1-z}\right) \tag{48}$$

$$\times \int dx \, d\phi \, dz' \frac{x^2 + 1 - x\cos\phi}{z'x\left(x^2 + 1 - 2x\cos\phi\right)} \Theta\left(\frac{z'x^2\left(1 + x^2 - 2x\cos\phi\right)}{z^3(1-z)} - \mathcal{L}\right) \Theta\left(\frac{z(1-z)R^2 p_\perp^2}{m_H^2} - x^2\right).$$

The integral over $z'$ can be done and the integral over the soft gluon kinematics becomes

$$\int dx \, d\phi \, dz' \frac{x^2 + 1 - x\cos\phi}{z'x\left(x^2 + 1 - 2x\cos\phi\right)} \Theta\left(\frac{z'x^2\left(1 + x^2 - 2x\cos\phi\right)}{z^3(1-z)} - \mathcal{L}\right) \tag{49}$$

$$= \int dx \, d\phi \frac{x^2 + 1 - x\cos\phi}{x\left(x^2 + 1 - 2x\cos\phi\right)} \log \frac{x^2\left(1 + x^2 - 2x\cos\phi\right)}{z^2(1-z)\mathcal{L}} \Theta\left(\frac{x^2\left(1 + x^2 - 2x\cos\phi\right)}{z^2(1-z)} - \mathcal{L}\right).$$

Now, we continue with approximation of this integral through examination of the upper and lower bounds on the angle $x$. The jet radius constraint sets the upper bound on $x$, while the observable $\mathcal{L}$ sets the lower bound. What we will do, then, is estimate this integral by taking its leading contribution in the limits that $x \to \infty$ or $x \to 0$, which we expect will at least describe the parametric scaling of the result. In the $x \to \infty$ limit, the integral becomes

$$\lim_{x\to\infty} \int dx \, d\phi \, [\cdot] \to \int \frac{dx}{x} d\phi \log \frac{x^4}{z^2(1-z)\mathcal{L}} \Theta\left(\frac{z(1-z)R^2 p_\perp^2}{m_H^2} - x^2\right) \tag{50}$$

$$\to \frac{\pi}{2} \log^2 \frac{R^2 p_\perp^2 \sqrt{1-z}}{m_H^2 \sqrt{\mathcal{L}}}.$$

In the $x \to 0$ limit, the integral instead becomes

$$\lim_{x\to 0} \int dx \, d\phi \, [\cdot] \to \int \frac{dx}{x} d\phi \log \frac{x^2}{z^2(1-z)\mathcal{L}} \Theta\left(x^2 - z^2(1-z)\mathcal{L}\right) \to 0. \tag{51}$$

With these asymptotics, the cumulative distribution is approximately

$$\Sigma_g(\mathcal{L}) \approx 1 - \frac{\alpha_s C_A}{2\pi} \frac{1}{2\log\frac{R^2 p_\perp^2}{m_H^2} - \frac{3}{2}} \int_{\frac{m_H^2}{R^2 p_\perp^2}}^{1/2} dz \left(\frac{1 + (1-z)^2}{z} + \frac{1+z^2}{1-z}\right) \log^2 \frac{R^2 p_\perp^2 \sqrt{1-z}}{m_H^2 \sqrt{\mathcal{L}}} \tag{52}$$

$$\approx 1 - \frac{\alpha_s C_A}{2\pi} \log^2 \frac{R^2 p_\perp^2}{m_H^2 \sqrt{\mathcal{L}}} + \mathcal{O}\left(\alpha_s \log \frac{R^2 p_\perp^2}{m_H^2 \sqrt{\mathcal{L}}}\right),$$

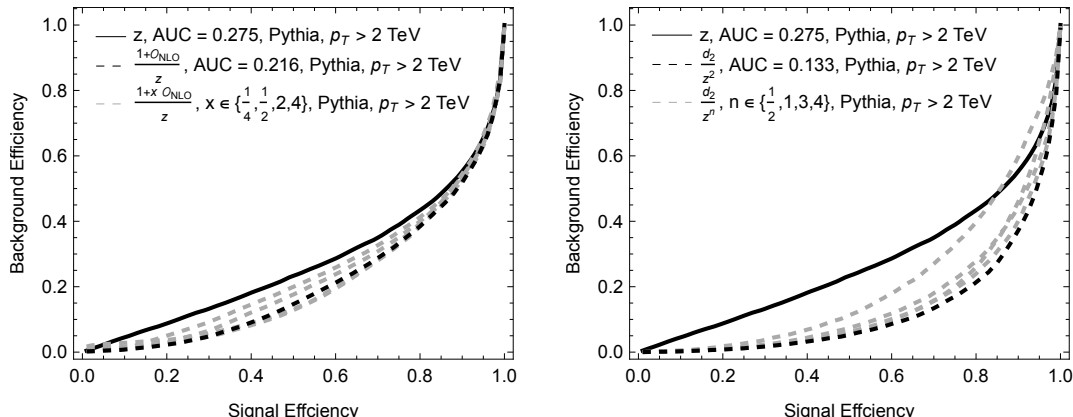

Figure 3: Plots of the ROC curves for optimal combinations of $\mathcal{O}_{\text{NLO}}$ (left) and $d_2$ (right) observables, with the soft subjet momentum fraction $z$ on simulated data in the jet transverse momentum bin $p_\perp > 2$ TeV. Also plotted are ROC curves for suboptimal observables, constructed by varying the linear combination parameter for $\mathcal{O}_{\text{NLO}}$ or the exponent of $z$ for $d_2$.

keeping only the leading double logarithms. Because this is only evaluated at next-to-leading order, it is only meaningful where the $\mathcal{O}(\alpha_s)$ term is less than 1, but nevertheless illustrates important scaling behavior. The background efficiency (or rejection rate) remains fixed as long as the ratio

$$\frac{R^2 p_\perp^2}{m_H^2 \sqrt{\mathcal{L}}} = \text{constant},\tag{53}$$

or that

$$\mathcal{L} = \frac{R^4 p_\perp^4}{m_H^4}.\tag{54}$$

Therefore, as $p_\perp \to \infty$, the value of the observable $\mathcal{L}$ at which the fixed background efficiency occurs increases rapidly. Further, the observable $\mathcal{L}$ is monotonically increasing with signal efficiency, and so, as $p_\perp$ grows with fixed background efficiency, the signal efficiency grows as well. Thus, in the high-boost limit, the observable $\mathcal{L} = d_2/z^2$ can have arbitrarily good signal efficiency with arbitrarily low background efficiency; or, the observable becomes the perfect discriminant in that limit.

## 5.2 Comparison with simulation

With these observables constructed and motivated from the structure of soft emissions in signal and background jets, we now move to implement them on our simulated data samples. In both observables $\mathcal{O}_{\text{NLO}}$ and $d_2$, angles between and from the direction of the two leading subjets are defined. As mentioned earlier, the directions of the two leading subjets are defined through reclustering the jet with the $k_T$ algorithm to 2 exclusive subjets, with Winner-Take-All recombination. Winner-Take-All recombination is vital because this ensures that the subjet directions are truly the direction of dominant energy flow, and are insensitive to recoil from soft, wide-angle emissions. Sensitivity to recoil has long been shown to reduce discrimination power in many jet classification problems [59].

With these observables, we plot their ROC curves on the highest jet transverse momentum sample in Fig. 3. On these plots, we compare the near-optimal "likelihood" observables on the spaces $(\mathcal{O}_{\text{NLO}}, z)$ and $(d_2, z)$ with just measuring the softer momentum fraction $z$ alone. Discrimination power is clearly improved with a more differential measurement. On these

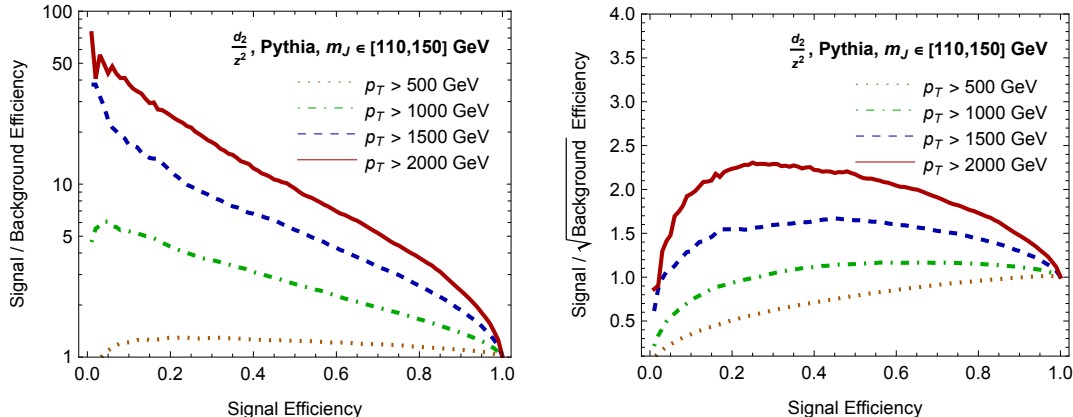

Figure 4: Plots of two quantifications of discrimination power with the observable $d_2/z^2$, in the four transverse momentum bins studied in this paper, $p_\perp > 500$ GeV (dotted), $p_\perp > 1000$ GeV (dot-dashed), $p_\perp > 1500$ GeV (dashed), and $p_\perp > 2000$ GeV (solid). Left: Signal to background efficiency ratio versus signal efficiency. Right: Signal to square-root background efficiency versus signal efficiency.

plots, we also display the ROC curves from a few different observables in which we vary a parameter in the functional form. With $\mathcal{O}_{\text{NLO}}$, we consider the observables

$$\mathcal{V}(x) \equiv \frac{1 + x\mathcal{O}_{\text{NLO}}}{z}\,, \tag{55}$$

varying the linear combination parameter $x \in \{1/4, 1/2, 2, 4\}$. The nominal parameter value, $x = 1$, does seem to perform the best over the entire range of efficiencies. With $d_2$, we consider the observables

$$\mathcal{D}(n) \equiv \frac{d_2}{z^n}\,, \tag{56}$$

varying the exponent parameter in $n \in \{1/2, 1, 3, 4\}$. The nominal selection $n = 2$, motivated through our theoretical calculations, is indeed the best discrimination observable for this parametrization of phase space. Further, we observe that $d_2/z^2$ is a significantly better discriminant than $(1 + \mathcal{O}_{\text{NLO}})/z$ as Fig. 3 demonstrates smaller background efficiency at fixed signal efficiency. Because of this, we will restrict further study to $d_2/z^2$ exclusively.

In Fig. 4, we then take the observable $d_2/z^2$ and display its discrimination power in a couple different ways, in all the jet transverse momentum bins studied in this paper. At left in this figure, we plot the signal versus background efficiency ratio as a function of the signal efficiency. At the highest transverse momenta, cuts on this observable can improve the $S/B$ ratio from its nominal value by nearly a factor of 30 or more, at reasonable signal rates. These plots already include a cut on the jet mass in the window $m_j \in [110, 150]$ GeV, and so the total signal versus background ratio improvement over inclusive jet cross sections can exceed 150, multiplying the ratios in this plot by the corresponding ratio of efficiencies listed in Table 1. At right in Fig. 4, we plot the improvement in significance, $S/\sqrt{B}$, from cuts on this observable, as a function of signal efficiency. In the highest jet transverse momentum bins, this can exceed 2, and, when the improvement from the mass cut is included, can exceed a factor of 5. This is still far from discovery potential, but perhaps a more detailed analysis of these jets can improve the picture further.

Note that at sufficiently low transverse momentum $p_\perp$, the discrimination power of $d_2/z^2$ is not much better than random guess. At lower $p_\perp$, especially close to the minimum at which both hard prongs from decay are captured in the jet, $p_{\perp,\text{min}} \sim 2m_H/R$, the hard subjets may be significantly sculpted by the jet algorithm. Further, at smaller $p_T$, the range of the softer

subjet's energy fraction $z$ decreases, and both effects reduce the efficacy of this observable for discrimination. Additionally, with sculpted subjets, their direction is smeared so that sensitivity to additional soft emissions through the observable $d_2$ is smeared, as $d_2$ is sensitive to the angles between soft emissions and the subjet directions. For these reasons, our simple perturbative calculations are most valid at the highest $p_T$ ranges.

## 6  Conclusions

We presented a systematic theoretical analysis of binary discrimination of $H \to gg$ decays from massive QCD jets through next-to-leading order in the high-boost limit. We validated our analytical results on simulated data, and observed that improvements to the signal-to-background ratio of several hundred were possible, even with only sensitivity to the first three dominant emissions in the jets. These results clearly and cleanly demonstrate the physics responsible for discrimination, and in the process produce observables that compactly encode this physics and can be employed in a realistic analysis very simply. This was guided by an anomaly-detection approach at next-to-leading order (and beyond), by which a powerful discrimination observable to distinguish signal from anything else is to define the observable to simply be the inverse of its matrix element. Such an observable is naturally IRC safe, because divergences are transformed into zeros by inversion, and so enables theoretical analysis and predictability.

This systematic approach suggests a way to validate that machine learning methods are indeed at the very least learning known, low-order, physics in their discrimination, and perhaps learning more detailed information as well. This is especially relevant in the context of the recent study of Ref. [60], in which it was explicitly demonstrated that the likelihood ratio for the problem of boosted top quark jets from QCD has rejection rates that are almost a factor of 10 larger than state-of-the-art machine learning architectures. The reason for this large gap is not currently known, whether there is genuine physics missing from the networks, or if the likelihood ratio on simulated data is so powerful because of effects or details of the simulation that are not generalizable, and so the networks should, and perhaps are, robust to. As a very first step to determining what is missing, it has not yet even been demonstrated that modern networks reproduce the likelihood for hadronc top decay versus QCD at leading-order, on jets with three resolved particles. This was recently constructed and analyzed in a similar method to the $H \to gg$ problem studied here [57], and will be central to demonstrating that the networks do indeed contain the known low-order physics and then go significantly beyond. Doing this will be an important step to interpretability of machine learning for particle physics applications.

## Acknowledgments

I thank Yonatan Kahn for continued encouragement to pursue this research.

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
