# Peer review of "Systematically Constructing the Likelihood for Boosted $H\to gg$ Decays"

_SciPost Physics, doi:SciPost Phys. 18, 130 (2025)_

## Round 1 · Referee Report · Anonymous (Referee 2) · 2025-3-12

Report

I thank the author for answering my questions and clarifying some points in the article. There is one point I would like to follow up on, please see my comment below.

Requested changes

On the previous point 9), the author's answer does not fully address my question. It's still not clear to me why the performance of the d2/z^2 observable is so much worse for low pT than for high pT. At a signal efficiency of about 0.2, the signal/background efficiency is about 1.1-1.2 for pt~500 GeV, but it is about 25 for pT~2000 GeV. Why is this variable not better than a random guess at low pT?

Connected to this, the author notes that Fig. 3 shows that d2/z^2 performs better than (1+O_NLO)/z and therefore only d2/z^2 is considered further. But Fig. 3 only shows the performance for pT>2 TeV. How does the comparison presented in Fig. 3 look at lower pT, or similarly, how does Fig. 4 look for (1+O_NLO)/z ? Can this information be added to the paper?

Recommendation

Ask for minor revision

  • validity: -
  • significance: -
  • originality: -
  • clarity: -
  • formatting: -
  • grammar: -

Author:  Andrew Larkoski  on 2025-03-24  [id 5309]

(in reply to Report 1 on 2025-03-12)

I thank the referee for their comments. At relatively low pT, around pT ~ 2m/R, the two hard prongs aren't necessarily completely captured in the jet. As such, the discrimination power of the softer subjet's energy fraction is a less powerful discriminant than at high pT. Further, if the hard prongs are less well-defined at lower pT, that has the consequence of making the soft dipole observable d_2 less efficient, as well. If the directions of the hard subjets are sculpted by the jet finding algorithm, then the angles of soft emissions from the hard subjets are smeared, and less correlated with the actual dipole that emitted them.

To address this, I have added the following paragraph to the end of section 5: "Note that at sufficiently low transverse momentum $p_\perp$, the discrimination power of $d_2/z^2$ is not much better than random guess. At lower $p_\perp$, especially close to the minimum at which both hard prongs from decay are captured in the jet, $p_{\perp,\min}\sim 2m_H/R$, the hard subjets may be significantly sculpted by the jet algorithm. Further, at smaller $p_T$, the range of the softer subjet's energy fraction $z$ decreases, and both effects reduce the efficacy of this observable for discrimination. Additionally, with sculpted subjets, their direction is smeared so that sensitivity to additional soft emissions through the observable $d_2$ is smeared, as $d_2$ is sensitive to the angles between soft emissions and the subjet directions. For these reasons, our simple perturbative calculations are most valid at the highest $p_T$ ranges."

Indeed, a comparison of the two NLO observables could be performed at lower pTs. However, with the perturbative calculations as a guide and as their validity decreases at lower pT, I felt that comparison of those distributions at lower pT was not necessary and somewhat distracting. That is, the goal of this paper is to use the perturbative analysis to understand the optimal observables where they are valid; i.e., at the highest pT. Then in Fig. 4, focusing on the observable d_2/z^2, this argument is validated by observation of significant reduction of discrimination power as pT decreases.

---

## Round 1 · Referee Report · Anonymous (Referee 1) · 2025-3-15

Report

The authors has addressed the points raised in my first report and in the one of a second referee. As far as I am concerned, I am satisfied with the answers and with the modifications made, and I confirm my recommendation of acceptance of this paper.

Recommendation

Publish (easily meets expectations and criteria for this Journal; among top 50%)

---

## Round 1 · Author Response

I have addressed the comments by the referees.

---

## Round 2 · Referee Report · Anonymous (Referee 2) · 2025-3-27

Report

I thank the author for answering my additional question and addressing my comment. Congratulations to this interesting and important study!

Recommendation

Publish (easily meets expectations and criteria for this Journal; among top 50%)

---

## Round 2 · Author Response

Updated manuscript to reflect referee comments

---

## Round 2 · List of Changes

See responses to referee reports

---

## Editorial Decision

published